# Serial Measurements of Circulating KL-6, SP-D, MMP-7, CA19-9, CA-125, CCL18, and Periostin in Patients with Idiopathic Pulmonary Fibrosis Receiving Antifibrotic Therapy: An Exploratory Study

**DOI:** 10.3390/jcm10173864

**Published:** 2021-08-28

**Authors:** Sebastian Majewski, Karolina Szewczyk, Aleksandra Żal, Adam J. Białas, Joanna Miłkowska-Dymanowska, Wojciech J. Piotrowski

**Affiliations:** 1Department of Pneumology, Medical University of Lodz, 90-153 Lodz, Poland; joanna.milkowska-dymanowska@umed.lodz.pl (J.M.-D.); wojciech.piotrowski@umed.lodz.pl (W.J.P.); 2Department of Pathobiology of Respiratory Diseases, Medical University of Lodz, 90-153 Lodz, Poland; szewczyk_karolina@wp.pl (K.S.); adam.bialas@umed.lodz.pl (A.J.B.); 3Department of General and Oncological Pulmonology, Medical University of Lodz, 90-549 Lodz, Poland; aleksandra.zal@umed.lodz.pl

**Keywords:** idiopathic pulmonary fibrosis, IPF, biomarker, KL-6, SP-D, MMP-7, CA19-9, CA-125, CCL18, periostin

## Abstract

Idiopathic pulmonary fibrosis (IPF) is a progressive and inevitably fatal disease with a heterogeneous clinical course. This study aimed to evaluate the usefulness of circulating biomarkers in routine IPF clinical practice. We conducted an exploratory study in a cohort of 28 IPF subjects qualified for anti-fibrotic therapy with up to 24 months serial measurements of seven IPF biomarkers, including those that are well-established, Krebs von den Lungen-6 (KL-6), surfactant protein D (SP-D), matrix metalloproteinase 7 (MMP-7), and more recently introduced ones, cancer antigen 19-9 (CA19-9), cancer antigen 125 (CA-125), chemokine (C-C motif) ligand 18 (CCL18), and periostin. Among studied biomarkers, SP-D had the highest diagnostic accuracy to differentiate IPF subjects from controls, followed by MMP-7 and KL-6. At each study timepoint, KL-6 levels correlated inversely with forced vital capacity % predicted (FVC% pred.), and transfer factor of the lung for carbon monoxide % predicted (T_L,CO_% pred.), while SP-D levels correlated inversely with FVC% pred. and T_L,CO_% pred. at 24 months of anti-fibrotic therapy. Baseline KL-6 and CA19-9 concentrations were significantly elevated in patients with progressive disease in comparison to patients with stable disease. In addition, in the progressors subgroup CA19-9 concentrations significantly increased over the second year of study follow-up. In patients with progressive disease, we observed a significant inverse correlation between a change in SP-D levels and a change in FVC% pred. in the first year of treatment, whereas in the second year a significant inverse correlation between a change in KL-6 levels and a change in FVC% pred. was noted. Our study findings support the view that both well-established IPF biomarkers, including KL-6, SP-D, and MMP-7, and more recently introduced ones, like CA19-9, have the potential to support clinical practice in IPF.

## 1. Introduction

Idiopathic pulmonary fibrosis (IPF) is a devastating and progressive lung disease with an inevitable fatal outcome. The clinical course of IPF is outlined by a progressive decline of lung function, physical activity limitation, impairment of quality of life, and premature death. The disease trajectories are variable and unpredictable and the median survival is 3 to 5 years [1,2], yet some patients live longer [3]. Chronic and repetitive micro-injuries to the alveolar epithelium caused by exposure to various noxious stimuli [4,5,6,7] lead in genetically predisposed individuals to subsequent dysfunction of the alveolar epithelial cells (AECs) which is central to the initiation and perpetuation of the pathogenic process in IPF. Aberrant activation of AECs leads to the production of mediators involved in the migration, proliferation, and activation of fibroblasts, their differentiation to myofibroblasts, and consequent excessive and chaotic secretion of extracellular matrix (ECM) proteins, leading to progressive lung parenchyma destruction characteristic for IPF [8].

In the last decade, the treatment landscape of IPF has changed significantly. Anti-fibrotic drugs have been shown to modify the disease course. Pirfenidone and nintedanib influence the disease progression limiting the decline of lung function in patients with IPF [9,10]. Both drugs are currently recognized as an actual standard of pharmacotherapy of IPF [11,12]. Despite accounting for about 20% of interstitial lung diseases (ILDs) [2], and holding a poor prognosis, IPF still faces considerable delays of early and reliable diagnosis and difficulties of accurate prognosis of disease behavior and response to treatment.

Biologic markers (biomarkers) are defined as objectively measured characteristics that can be evaluated as indicators of normal biologic processes, pathogenic processes, or pharmacologic responses to therapeutic intervention [13]. A large and growing body of literature has investigated changes in the level of various potential diagnostic and prognostic biomarkers in patients with IPF which could serve as a new tool for clinical practice in IPF. Many clinical studies have identified different circulating biomarkers associated with AECs dysfunction, ECM remodeling, and fibroproliferation or have been associated with immune dysfunction, reflecting various disease aspects in IPF [14,15]. Among them there are well-established biomarkers, extensively studied in IPF, including Krebs von den Lungen-6 (KL-6), surfactant protein D (SP-D), matrix metalloproteinase 7 (MMP-7), and more recently introduced biomarkers, less extensively studied, including cancer antigen 19-9 (CA19-9), cancer antigen 125 (CA-125), chemokine (C-C motif) ligand 18 (CCL18) and periostin. To date, only a few studies investigated the utility of different circulating biomarkers for clinical practice in patients with IPF receiving anti-fibrotic treatment [16,17]. In addition, it is not clear what impact anti-fibrotic treatment has on the expression of circulating biomarker levels, and whether potential biomarkers could help us to monitor how effective treatment for IPF is.

The present study aimed to evaluate the usefulness of circulating biomarkers in routine IPF clinical practice. We conducted an exploratory, longitudinal study with up to 24 months of serial measurements of seven IPF serum biomarkers, including KL-6, SP-D, MMP-7, CA19-9, CA-125, CCL18, and periostin in a cohort of IPF patients qualified for anti-fibrotic therapy at a single reference center. Our study findings support the view that both well-established IPF biomarkers, including KL-6, SP-D, and MMP-7, and more recently introduced ones, like CA19-9, have the potential to support clinical practice in IPF.

## 2. Materials and Methods

### 2.1. Study Population

A total of 48 volunteers participated in this study, including 28 treatment-naïve patients with IPF qualified for anti-fibrotic therapy and 20 controls. IPF diagnosis was established according to the international guidelines [1], whereas control subjects had neither a history nor symptoms of lung diseases. The IPF cohort included 21 patients treated with pirfenidone and 7 patients treated with nintedanib for up to 24 months. The study protocol was reviewed and approved by the Ethics Committee of the Medical University of Lodz (approval number RNN/66/17/KE, date 14 March 2017). The study was conducted according to the Declaration of Helsinki’s principles. Before the start of any study procedures, all participants gave written informed consent for their participation.

### 2.2. Methods

All study participants underwent clinical assessments, including medical history, physical examination, peripheral venous blood sampling, and pulmonary function tests (PFTs). Controls were subjected to baseline blood sampling and spirometry assessment, whereas patients with IPF underwent peripheral venous blood sampling, spirometry, single-breath transfer factor of the lung for carbon monoxide (T_L,CO_) measurement, and a six-minute walk test (6MWT) at baseline and then after 6, 12, 18 and 24 months of anti-fibrotic therapy. Concentrations of serum biomarkers were evaluated by enzyme-linked immunosorbent assay (ELISA).

### 2.3. Pulmonary Function Assessments

Spirometry and T_L,CO_ measurements were performed using the Lungtest 1000 system (MES, Cracow, Poland) according to ATS/ERS standards [18,19]. Forced expiratory volume in 1 s (FEV_1_), forced vital capacity (FVC), FEV_1_/FVC%, and T_L,CO_ corrected for hemoglobin concentration were recorded. For the expression of PFT results as percent of the predicted values (% pred.), we adopted the Global Lung Function Initiative (GLI) reference values throughout the study timepoints [20,21].

### 2.4. The Six-Minute Walk Test (6MWT)

The 6MWT is a practical and reliable measure of functional exercise capacity in patients with different cardiac and respiratory diseases, including IPF [22]. In the present study, 6MWT was performed according to the methodology specified by the Polish Respiratory Society guidelines [23]. Briefly, study participants were instructed to walk as far as possible for 6 min. The 6MWT was performed indoors, on a flat, straight corridor that was 30 m long and marked meter by meter. When the 6MWT was finished, the distance covered was calculated and recorded.

### 2.5. Composite Physiologic Index (CPI) and Gender-Age-Physiology (GAP) Index

CPI was developed as a tool to reflect the morphologic extent of fibrosis in patients with IPF [24]. It is calculated as follows: 91.0 − (0.65 × T_L,CO_% pred.) − (0.53 × FVC% pred.) + (0.34 × FEV_1_% pred.). GAP index and staging system provide an estimation of the average mortality risk of IPF patients by GAP stage [25]. It combines clinical variables, including age and gender, as well as physiological variables, including FVC and T_L,CO_. Both CPI and GAP have been shown to provide prognostic information in patients with IPF.

### 2.6. Serum Biomarkers Measurements

Peripheral venous blood samples were drawn into serum separator tubes (SST) (BD Dickinson, Franklin Lakes, NJ, USA). Samples were left for 30 min at room temperature for blood clotting. After that time, samples were centrifuged for 15 min at 1000× *g* and stored at −80 °C for further assessments. Serum biomarkers levels were measured in duplicate using commercially available enzyme-linked immunosorbent assays: KL-6 (Sekisui Medical Co., Ltd., Tokyo, Japan), SP-D (BioVendor-Laboratorni medicina a.s., Brno, Czech Republic), MMP-7 (Biorbyt Ltd., Cambridge, UK), CA19-9 (Dia.Metra S.r.I., Spello, Italy), CA-125 (Dia.Metra S.r.I., Spello, Italy), CCL18 (RayBiotech, Norcross, GA, USA) and total periostin (RayBiotech, Norcross, GA, USA), according to the manufacturer’s instructions.

### 2.7. Definition of Idiopathic Pulmonary Fibrosis (IPF) Progression

IPF progression was assessed based on the complex analysis of changes in physiologic (FVC and T_L,CO_) and functional measures (6MWT) after 12 months and 24 months since anti-fibrotics initiation. The composite definition of disease progression was described as ≥10% absolute decline in FVC% pred. and/or ≥15% absolute decline in T_L,CO_% pred. and/or ≥50 m decline in 6MWT distance within 12 months of therapy (12 months vs. baseline and 24 months vs. 12 months’ study timepoint). Patients with disease progression were classified into the progressors subgroup, whereas the others were classified into the stables subgroup.

### 2.8. Analysis of Associations between Longitudinal Changes in Serum Biomarkers Levels with Changes in Physiologic and Functional Measures in Patients with IPF

All changes were evaluated separately in the subgroup of stables and progressors in the first and second year of anti-fibrotic therapy. Changes in serum biomarkers levels and 6MWT distance were calculated as relative change and expressed in %, whereas, the changes in FVC and T_L,CO_ were expressed as the absolute change of % predicted values.

### 2.9. Statistical Analysis

The study data were analyzed using GraphPad Prism 8 (GraphPad Software, La Jolla, San Diego, CA, USA) except for the receiver-operating characteristic (ROC) curves analysis for which IBM SPSS Statistics version 27.0 (IBM, Armonk, NY, USA) was used. The normality of data distribution was tested with the Shapiro–Wilk test. Continuous data are expressed as mean with standard deviation (SD) for normally distributed data or as median with interquartile range (IQR) for non-parametric data. Categorical data are presented as absolute numbers and relative frequencies. Data were analyzed using unpaired *t*-test, the Mann–Whitney U test, paired *t*-test, or Wilcoxon signed-rank test depending on data normality and homogeneity of variance. To compare multiple groups of the paired sample we applied the Friedman test with Dunn’s correction for non-parametric data, whereas for parametric data one-way repeated measures ANOVA with the Geisser–Greenhouse correction was used. The Spearman correlation coefficient was applied to evaluate associations between variables. The area under the ROC curve (AUC) analysis was used to evaluate the discriminating capability of circulating biomarkers to differentiate IPF subjects from controls. Cut-off levels for serum biomarkers concentrations were determined using the Youden’s J statistic, which maximizes sensitivity and specificity. Statistical significance was accepted at *p* < 0.05.

## 3. Results

### 3.1. Patients’ Characteristics

We included 28 patients with IPF and 20 control subjects. In the IPF cohort, subjects were mostly male (60%) with a mean age of 69.1 (7.8) years. There was no significant difference in age between IPF subjects and controls. Baseline PFTs data of patients with IPF showed mild to moderate lung function impairment. The majority of IPF subjects were classified as stage I according to the GAP index and none of the patients with advanced disease classified as GAP stage III were enrolled in this study. The baseline characteristics of study participants are presented in Table 1.

### 3.2. Baseline Serum Biomarkers Concentrations and Receiver-Operating Characteristic (ROC) Curve Analysis Discriminating IPF from Control Subjects

Baseline serum KL-6, SP-D, MMP-7, CA-125, CCL18, and periostin levels were measurable in all studied subjects, whereas serum CA19-9 levels were measurable in 46 out of 48 participants. Study results showed significantly elevated concentrations of all circulating biomarkers studied (all *p* < 0.05), except for periostin, in patients with IPF compared with controls, see details in Table 1 and graphical presentation in Figure 1. The ROC curve analysis indicated that SP-D had the highest diagnostic accuracy in distinguishing IPF from control subjects (AUC 0.9089, 95% CI 0.8196–0.9983, *p* < 0.0001) followed by MMP-7 (AUC 0.8946, 95% CI 0.8011–0.9882, *p* < 0.0001), and KL-6 (AUC 0.8589, 95% CI 0.7430–0.9749, *p* < 0.0001). At the cut-off level of ≥121.8 ng/mL, serum SP-D yielded a sensitivity of 100% and specificity of 75% to distinguish IPF from control subjects. More detailed results including the optimal discriminatory cut-off values and individual discriminatory specificity and sensitivity for each of the biomarkers studied are presented in Figure 2 and Table 2.

### 3.3. Longitudinal Associations between Serum Biomarkers Levels in Patients with IPF during Antifibrotic Therapy

We noted negative correlations between circulating SP-D levels and CA-125 levels at each study timepoint during anti-fibrotic therapy of IPF subjects: at baseline r = −0.52, *p* < 0.01, at 6 months r = −0.53, *p* < 0.01, at 12 months r = −0.45, *p* < 0.05, at 18 months r = −0.48, *p* < 0.05 and at 24 months r = −0.49, *p* < 0.01. At baseline, we also found a significant positive correlation between serum KL-6 levels and serum SP-D levels (r = 0.49 *p* < 0.01) as well as between serum KL-6 levels and serum CCL18 levels (r = 0.44, *p* < 0.05). Similarly, serum KL-6 levels correlated positively with serum SP-D levels (r = 0.46, *p* < 0.05) at 6 months of treatment. Moreover, at 12 months of treatment, we observed a positive correlation between serum CCL18 levels and serum CA-125 levels (r = 0.45, *p* < 0.05), whereas at 18 months of treatment serum CCL18 levels correlated negatively with serum MMP-7 levels (r = −0.41, *p* < 0.05). All possible associations between circulating biomarkers over 24 months of anti-fibrotic therapy in patients with IPF are presented in Appendix A.

### 3.4. Longitudinal Associations of Serum Biomarkers Levels with Clinical Measures in Patients with IPF during Antifibrotic Therapy

The longitudinal changes in PFTs, 6MWT, and circulating biomarkers concentrations were evaluated in 6-month intervals. Over a study follow-up, the mean FVC in our cohort was relatively preserved, while the mean T_L,CO_ was steadily decreasing. The mean distance covered in the 6MWT also decreased significantly after 24 months of study duration. No significant differences between the mean levels of circulating biomarkers measured in consecutive study timepoints were noted. Longitudinal changes in PFTs, 6MWT, and circulating biomarkers concentrations in a cohort of patients with IPF are shown in Figure 3 and Figure 4, more detailed data are presented in Appendix A.

Analysis of possible associations between circulating biomarker concentrations and clinical measures in patients with IPF revealed negative correlations between serum KL-6 levels and FVC% pred. at each study timepoint: at baseline r = −0.67, *p* < 0.001, at 6 months r = −0.57, *p* < 0.01, at 12 months r = −0.60, *p* < 0.001, at 18 months r = −0.41, *p* < 0.05 and at 24 months r = −0.50, *p* < 0.01. Similarly, serum KL-6 levels correlated inversely with T_L,CO_% pred. during a study follow up: at baseline r = −0.53, *p* < 0.01, at 6 months r = −0.46, *p* < 0.05, at 12 months r = −0.48, *p* < 0.01, at 18 months r = −0.53, *p* < 0.01 and at 24 months r = −0.63, *p* < 0.001. At 18 months of anti-fibrotic treatment serum KL-6 levels correlated positively with CPI score (r = 0.46, *p* < 0.05), whereas, at 24 months serum KL-6 levels correlated negatively with age (r = −0.42, *p* < 0.05). At 24 months, we found significant negative correlations between serum SP-D levels and FVC% pred. (r = −0.41 *p* < 0.05) and T_L,CO_% pred. (r = −0.47, *p* < 0.05). Additionally, we noted a significant positive correlation between serum SP-D levels and GAP index (r = 0.43, *p* < 0.05) after 24 months of therapy. Significant positive correlations were observed at baseline, 6, 18 and 24 months of study duration between serum CA19-9 levels and time since diagnosis (r = 0.45, *p* < 0.05; r = 0.39, *p* < 0.05; r = 0.42, *p* < 0.05; r = 0.45, *p* < 0.05, respectively). Similarly, serum CA19-9 levels correlated positively with CPI score at 6 months (r = 0.41, *p* < 0.05), at 12 months (r = 0.39, *p* < 0.05) and at 24 months (r = 0.56, *p* < 0.01) of anti-fibrotic treatment. We also observed inverse correlations between serum CA19-9 concentrations and T_L,CO_% pred. at 6 months (r = −0.39, *p* < 0.05) and at 24 months (r = −0.48, *p* < 0.05) of the study follow up. Moreover, at 6 months timepoint we observed a positive correlation between serum CA-125 levels and CPI score (r = 0.50, *p* < 0.01). We noted a positive correlation between serum CCL18 concentrations and CPI score at baseline r = 0.44, *p* < 0.05, at 6 months r = 0.45, *p* < 0.05, at 12 months r = 0.42, *p* < 0.05, and at 24 months r = 0.40, *p* < 0.05. Serum CCL18 levels were inversely correlated with T_L,CO_% pred. at baseline r = −0.38, *p* < 0.01, and at 24 months r = −0.48, *p* < 0.05. In addition, serum CCL18 concentrations correlated positively with the GAP index (r = 0.42, *p* < 0.05) at 24 months of treatment. All possible correlations of circulating biomarkers concentrations with clinical measures are presented in Appendix A.

### 3.5. Dynamic Changes in PFTs, 6MWT, and Serum Biomarkers Levels According to the Disease Progression Assessment of Patients with IPF

Based on the criteria used for the composite definition of IPF progression, in the first year of anti-fibrotic therapy 10 subjects were classified as progressors and 18 subjects were classified as stables, whereas in the second year 11 subjects were classified as progressors and 17 subjects were classified as stables. None of the patients experienced progression consecutively during both 12-month periods, therefore different patients were classified into the progressors’ subgroup in the first and second year study follow ups. In the first year, the progressors subgroup included 2 patients on nintedanib and 8 patients on pirfenidone therapy, whereas in the second year, 5 patients on nintedanib and 6 patients on pirfenidone.

Longitudinal changes in PFTs and 6MWT in IPF patients with stable and progressive disease are presented in Figure 5. Over a study follow-up, the mean FVC% pred. in the subgroup of stables and progressors did not differ significantly between consecutive study timepoints over the first and the second 12-month periods, while the mean T_L,CO_% pred. decreased significantly in both subgroups of patients in the first year study follow-up. The mean distance covered in the 6MWT decreased significantly in the progressors subgroups of patients over the first and the second year of study duration.

Analysis of serum biomarkers levels in subgroups of patients with progressive and stable disease within the first year of anti-fibrotic therapy revealed significantly increased baseline serum KL-6 concentrations in the progressors subgroup compared with the stables subgroup (1485 (1182–3743) U/mL vs 965.5 (581.8–1533) U/mL; *p* < 0.05). In addition, baseline serum CA19-9 concentrations were also significantly elevated in patients with progressive disease compared with patients with stable disease categorized based on the first year of treatment outcomes (25 (14.85–26.53) vs 12.4 (6.85–21) U/mL; *p* < 0.05), see Figure 6. No similar significant differences in any of the biomarkers studied were observed between the stables and progressors during the second year of anti-fibrotic therapy (analyzed at 12-month study timepoint respectively), however, we noted increased serum SP-D levels in patients with progressive disease compared with patients with stable disease at 24 months (494.5 (297.3–640.1) vs 271.9 (188.2–470.7 ng/mL; *p* < 0.05), see Figure 7.

Additional analysis of dynamic changes in circulating biomarkers concentrations in subgroups of patients performed in the first year of treatment revealed significantly increased serum KL-6 levels at 12 months in comparison with baseline in patients with stable disease (1393 (594.8–2173) U/mL vs 965.5 (581.8–1533) U/mL; *p* < 0.05), see Figure 6. Morover, in the second year of anti-fibrotic therapy, serum CA19-9 levels were significantly increased at 24 months compared to 12 months in patients with progressive disease (22.00 (14.30–35.40) U/mL vs 15.00 (11.20–28.90) U/mL; *p* < 0.01), see Figure 7. No significant dynamic changes of any other biomarker studied according to the disease progression assessment were noted. No additional analyses were performed accounting for different anti-fibrotics used due to a small number of subjects within each subgroup of patients.

The ROC curve analysis indicated that baseline serum KL-6 had slightly higher diagnostic accuracy than CA19-9 in distinguishing patients with a stable course of disease from patients with a progressive course of disease over the first year of anti-fibrotic therapy (AUC 0.7417, 95% CI 0.5535–0.9298, *p* < 0.05). At the cut-off level of ≥1328 U/mL, serum KL-6 yielded a sensitivity of 70% and specificity of 72% to discriminate stables from progressors. Detailed results of ROC curve analysis in circulating biomarkers to distinguish stables from progressors are presented in Figure 8 and Table 3.

### 3.6. Associations of Longitudinal Changes in Serum Biomarkers Levels and Changes in Physiologic and Functional Measures in Subgroups of Patients with IPF

In the stables subgroup of patients, no significant relationships between changes in serum biomarkers concentrations and changes in physiological or functional measures were noted over the study follow-up. In the progressors subgroup, in the first year of treatment, we observed a significant strong inverse correlation between a change in serum SP-D levels and a change in FVC% pred. (r = −0.72, *p* < 0.05), whereas in the second year, we noted a significant strong inverse correlation between a change in serum KL-6 levels and a change in FVC% pred. (r = −0.71, *p* < 0.05). A significant strong positive association was observed between a change in serum CCL18 levels and a change in 6MWT distance in the progressors subgroup of patients over the first year of anti-fibrotic therapy, see Figure 9. All possible relationships of changes in circulating biomarkers levels with physiological and functional measures in patients with IPF are presented in Appendix A.

## 4. Discussion

In the present study, we directly compared the clinical usefulness of seven circulating biomarkers, including KL-6, SP-D, MMP-7, CA19-9, CA-125, CCL18, and periostin, serially measured in a cohort of patients with IPF receiving anti-fibrotic therapy over 24 months. Baseline ROC curve analysis showed that serum SP-D, MMP-7, and KL-6 had the highest diagnostic accuracy in distinguishing IPF from control subjects, which confirms their potential as diagnostic biomarkers. No significant differences between the mean concentrations of each serially measured biomarker in the consecutive study timepoints were noted. This finding suggests that stable serum biomarker concentrations could be associated with treatment response. The longitudinal data revealed inverse correlations of serum KL-6 levels with FVC% pred. and T_L,CO_% pred. at each study timepoint, and serum SP-D levels with FVC% pred. and T_L,CO_% pred. at 24 months study timepoint. Baseline serum KL-6 and CA19-9 concentrations were most clearly distinguishing patients with progressive and stable disease. Moreover, in patients with progressive disease, serum concentrations of CA19-9 significantly increased over the second year of study follow-up. As far as we know, no previous evidence has shown a dynamic increase of serum CA19-9 in IPF patients with progressive disease during anti-fibrotic therapy. In addition, in the progressors subgroup, significant strong inverse correlations were noted between a change in serum SP-D levels and a change in FVC% pred. in the first year of anti-fibrotic therapy, and between a change in serum KL-6 levels and a change in FVC% pred. during the second year of anti-fibrotic therapy. These results add further evidence supporting the potential role of KL-6, SP-D, and CA19-9 as prognostic biomarkers in IPF. Taken together, our research findings support the view that both well-established IPF biomarkers, including KL-6, SP-D, and MMP-7, and more recently introduced ones, like CA19-9, have the potential to support clinical practice in IPF.

The differential diagnosis of IPF may be challenging. Thus, a combination of clinical, laboratory, imaging, and in selected cases lung biopsy data are required for diagnosing IPF [1,12]. The present study results indicate that serum SP-D, MMP-7, and KL-6 could serve as diagnostic biomarkers of IPF which could improve early diagnosis. The ROC curve analysis showed accomplished discriminatory capability for these three biomarkers as indicated by the high AUC value (0.85 to 0.9). In our study, baseline SP-D had the highest diagnostic accuracy (AUC 0.9) in distinguishing IPF from control subjects. Surfactant proteins are synthesized and secreted by type 2 AECs and facilitate the transport and function of surfactant lipids. A systematic review and meta-analysis confirm that detection of serum SP-D might be useful for differential diagnosis in patients with IPF [26]. It is of note that serum SP-D levels were significantly higher in patients with IPF than in patients with pulmonary infection and healthy individuals, but no significant difference was found in serum SP-D concentrations between patients with IPF and those with non-IPF ILDs [26]. Matrix metalloproteases (MMPs) are endopeptidases, capable of degrading ECM proteins. Timely degradation of ECM is an important feature of various biologic processes including development, morphogenesis, tissue repair, and remodeling. It has been shown that serum MMP-7 levels are significantly elevated in patients with IPF compared with healthy controls [27,28,29,30,31,32] and lung diseases other than ILDs [27] and non-IPF ILDs [27,28]. KL-6 is a mucin-like, high-molecular-weight glycoprotein expressed on the surface membrane of AECs. KL-6 is released into the systemic circulation when AECs proliferate, are activated, or injured. Previous studies showed that serum KL-6 levels are significantly elevated in patients with IPF compared to healthy volunteers [17,29,33,34,35,36] and patients with bacterial pneumonia or other ILDs [29,33]. It was proposed that a cut-off value of 465 U/mL of serum KL-6 can distinguish subjects with ILDs from healthy individuals and patients with lung diseases other than ILDs [33]. However, KL-6 serum levels higher than the cut-off value have been observed in more than 70% of patients with various ILDs which points out the limited clinical utility of serum KL-6 levels in the differential diagnosis of ILDs [37].

Tumor markers, CA19-9 and CA-125, are secreted in small amounts by pulmonary epithelium in states of health, but are secreted in abundance by metaplastic epithelium in patients with IPF and are considered markers of epithelial damage. Our data fit well with the previous studies showing their elevated serum levels in IPF, non-IPF ILDs, and other than ILDs respiratory disorders [38,39], however, their role in the differential diagnosis of IPF is limited. Surprisingly, we noted inverse correlations between circulating SP-D levels and CA-125 levels at each study timepoint. Both SP-D and CA-125 are considered epithelial biomarkers. To our knowledge, such an observation has not been reported previously. Further studies in larger IPF cohorts including studies analyzing possible mechanistic link for this observation are needed to validate this finding. CCL18 is produced primarily by antigen-presenting cells, including macrophages, dendritic cells, and peripheral blood monocytes. It is a small protein that acts as a chemoattractant involved in immune cell trafficking. Our results of increased serum CCL18 levels in IPF are in line with the previous studies showing increased levels of CCL18 in various ILDs, including IPF, as compared with healthy controls [29,40]. Periostin is an ECM protein that contributes to mesenchymal cell proliferation and fibrosis in different organs including the lung and can be secreted by bronchial epithelial cells in response to interleukin-13, which was found to be involved in the development and progression of IPF [41]. Unfortunately, our study has not confirmed previously reported elevated periostin levels in patients with IPF compared to controls and its diagnostic and prognostic utility in IPF [42,43]. It has been shown that monomeric periostin levels can be used to distinguish IPF patients from healthy controls and predict pulmonary function decline [43]. Noted discrepancies between our and other data could result from the use of conventional ELISA kit detecting total periostin including both monomeric and oligomeric periostin in our research and not new ELISA kit specifically detecting its monomeric form [43].

Taking together the above considerations, our results provide additional support for the potential value of serum SP-D, MMP-7, and KL-6 concentrations as diagnostic biomarkers in IPF. This finding is in good agreement with the recent PROFILE (Prospective Observation of Fibrosis in the Lung Clinical Endpoints) study results indicating that serum SP-D and MMP-7 could best discriminate between IPF patients and controls based on the analysis of 44 proteins differentially expressed in a large IPF cohort and control subjects. It is of note that the PROFILE study represents the largest prospective analysis of serum biomarkers in IPF to date. Nevertheless, the actual clinical utility of diagnostic biomarkers in IPF is underscored by the lack of reproducible, uniform cut-off values allowing differentiation of IPF from healthy controls, lung diseases other than ILDs, and various non-IPF ILDs. As for now, measurement of circulating IPF biomarkers offers only a potential supportive role in improving the accuracy of differential diagnosis of IPF.

In our study population, the mean decrease of FVC observed after 24 months of anti-fibrotic therapy was only 90 mL. In addition, the average concentrations of circulating biomarkers remained largely stable and no significant differences between consecutive study timepoints were noted. These findings appear to be well supported by the previous studies demonstrating stable levels of circulating KL-6 during anti-fibrotic therapy with pirfenidone or nintedanib which is in good agreement with our study results [35,44]. We hypothesize that the above findings may be indicative of a good response to anti-fibrotic therapy observed in our cohort of patients with IPF. Therefore, our and other data suggest that stable serum biomarkers concentrations could be associated with treatment response but this needs further longitudinal studies in larger cohorts of patients.

The longitudinal analysis of possible associations of circulating biomarkers with clinical measures in IPF revealed inverse correlations of serum KL-6 and SP-D concentrations with serially measured FVC% pred. and T_L,CO_% pred. in our cohort. These results substantiate previous findings in the literature of associations between serum KL-6 and SP-D concentrations and longitudinally measured FVC and T_L,CO_ in patients with IPF [45,46,47]. To further assess possible associations between changes of serum biomarkers concentrations and disease progression we have used a composite definition of IPF progression, combining assessment of physiologic (FVC and T_L,CO_), and functional (6MWT) markers of disease severity. An absolute decline in FVC of ≥10% or T_L,CO_ of ≥15% over 6 to 12 months has been regarded as clinically important and is frequently used to describe a significant disease progression [48]. The substantial evidence from independent retrospective and prospective studies demonstrates that ≥10% decline in FVC is associated with a significant increase in mortality in IPF [49,50,51,52]. Data regarding changes in T_L,CO_ as a predictor of outcomes in IPF are inconclusive, nevertheless, studies show that trends in T_L,CO_ levels might provide important information for determining mortality [51]. The longitudinal variation in the 6MWT distance has been used to reflect the disease status and progression [53,54,55], prognosis prediction [56] and has been shown to outweigh other predictors of mortality in IPF [57]. A threshold for minimum clinically important difference value for 6MWT has been suggested as 24–45 m or more [58]. It is likely that such a combined definition of disease progression, including physiological and functional measures, reflects more broadly clinically significant deterioration of patients with IPF in routine clinical practice. Based on our study results, the biomarkers that most clearly distinguished patients with progressive and stable disease were serum KL-6 and CA19-9 for which baseline concentrations were substantially raised in the progressors subgroup compared with the stables subgroup. The ROC curve analysis showed similar discriminatory capability for both biomarkers as indicated by the AUC value (0.7417 for KL-6 and 0.7306 for CA19-9). The previous studies have shown that serum KL-6 levels at diagnosis are predictive of IPF progression [59,60] and that serum KL-6˃1000 U/mL and increasing serum KL-6 concentrations over time are associated with worse outcomes and higher risk of mortality in IPF [17,60,61,62]. It is of note that the median baseline serum KL-6 level in the progressors subgroup of patients in our study was 1485 U/mL. Similar to our data, high baseline serum CA19-9 levels were able to predict the 12-month progression of patients with IPF in the PROFILE study [32]. Surprisingly, our analysis of serial changes in biomarker concentrations revealed significantly increased serum KL-6 levels in the stables subgroup of patients after 12 months of anti-fibrotic treatment compared with baseline. It is plausible that the reason for this unexpected finding and evident discrepancy with the literature data is a small sample size studied which could have influenced the results obtained. Among all longitudinally measured biomarkers in our study, only serum CA19-9 concentrations significantly increased in the progressors subgroup of IPF patients over the second year of anti-fibrotic therapy. To our knowledge, no previous evidence has shown a dynamic increase of serum CA19-9 in IPF patients with progressive disease during anti-fibrotic treatment. This result appears well supported by our and PROFILE study findings that serum CA19-9 could distinguish patients with progressive disease from patients with stable disease [32]. However, no significant dynamic change in 3-months interval was noted for the serum CA19-9 concentrations in the progressors subgroup of patients in the PROFILE study. In contrast to the PROFILE study results, we did not confirm rising concentrations of serum CA-125 in patients with progressive disease [32]. The foremost cause of discrepancies between our study data and the PROFILE study data may result from differences in the definition of disease progression, which in the PROFILE study was defined as all-cause mortality or a decline of ≥10% in FVC within 12 months since baseline, and a shorter interval of serum biomarkers measurements in the progressors’ subgroup of patients (12 months vs. 3 months) [32]. It is noteworthy that serum CA19-9 concentrations were associated with disease duration (time since diagnosis) and CPI scores in serial measurements performed in our study. In our view, these novel findings suggest that increasing serum CA19-9 levels may reflect an accelerating extent of epithelial damage and fibrosis observed in patients with IPF longitudinally, with higher levels noted in patients with progressive disease. Thus, our and other data lend support for circulating CA19-9 as a novel and promising biomarker of progression in IPF. Another novel observation revealed in our study is the positive association between dynamic changes of CCL18 and changes of 6MWT distance over the first 12 months of anti-fibrotic treatment. No previous studies reported a similar finding, and no clear explanation for such an observation arises from the literature. Thus, further studies are warranted to validate and possibly explain this exploratory finding. The current study does not confirm previous findings supporting the prognostic role of serum MMP-7, and CCL18 measured at the time of diagnosis as predictors of lung function decline and disease progression in IPF [16,31,45,63,64]. However, we found longitudinal positive associations of serum CCL18 concentrations with the CPI scores and a relationship between serum CCL18 concentration with the GAP index after 24 months of anti-fibrotic therapy which suggests that serum CCL18 expression may reflect disease advancement and severity. In addition, our study results support previous findings in the literature demonstrating relationships of longitudinal changes of serum SP-D and KL-6 concentrations with longitudinal changes in FVC% pred. in IPF patients experiencing progressive disease trajectory [46,62]. These results substantiate available evidence supporting the potential prognostic role of serially measured SP-D and KL-6 to assess disease prognosis in IPF.

In summary, apart from the slight discordance in our results, we believe that our study data corroborate well with the previous research in the field. There is substantial evidence that baseline, as well as dynamic changes in serum biomarkers concentrations, which correlate to disease progression, could be useful for assessing the prognosis in IPF.

Our study findings must be considered in the context of several limitations. First, the sample size is relatively small which might lead to either over- or underestimation of certain effects. Secondly, a selection bias due to exclusion of patients with insufficient measurements for evaluation of patterns in serial changes of biomarkers combined with clinical measures, and exclusion of patients with more advanced IPF could have influenced the present study results. Patients with more advanced IPF (FVC <50% of pred. and T_L,CO_ <30% of pred.) are not eligible to receive anti-fibrotics in the frame of reimbursed IPF treatment program in our country, therefore, they could not be enrolled into the present study. Thirdly, our study has not included survival analysis, which potentially limits the evaluation of the prognostic ability of studied biomarkers. Further longitudinal studies in larger patient cohorts are warranted to validate the present study findings.

## 5. Conclusions

Our study data showed that serum SP-D, MMP-7, and KL-6 could differentiate IPF from control subjects with high diagnostic accuracy. These results indicate that the measurements of serum concentrations of SP-D, MMP-7, and KL-6 may offer a potential tool to support an early diagnosis of IPF. We have also shown that baseline KL-6 and CA19-9 levels can discriminate between patients with progressive and stable disease. In addition, for the first time, we have demonstrated a dynamic increase of serum CA19-9 levels in IPF patients with progressive disease during anti-fibrotic treatment. Furthermore, we have confirmed that longitudinally measured changes in serum KL-6 and SP-D concentrations inversely correlate with the changes of lung function in the progressors subgroup of patients with IPF. These results add further evidence supporting the potential role of KL-6, SP-D, and CA19-9 as prognostic biomarkers in IPF. Taken together, the present study findings support the view that both well-established circulating IPF biomarkers, including KL-6, SP-D, and MMP-7, and more recently introduced ones, like CA19-9, have the potential to support clinical practice in IPF. Further work needs to be done in larger cohorts of patients to confirm exploratory findings of our research and identify the most useful diagnostic and prognostic biomarkers which could find their way into routine IPF clinical practice in the near future.

## Figures and Tables

**Figure 1 jcm-10-03864-f001:**
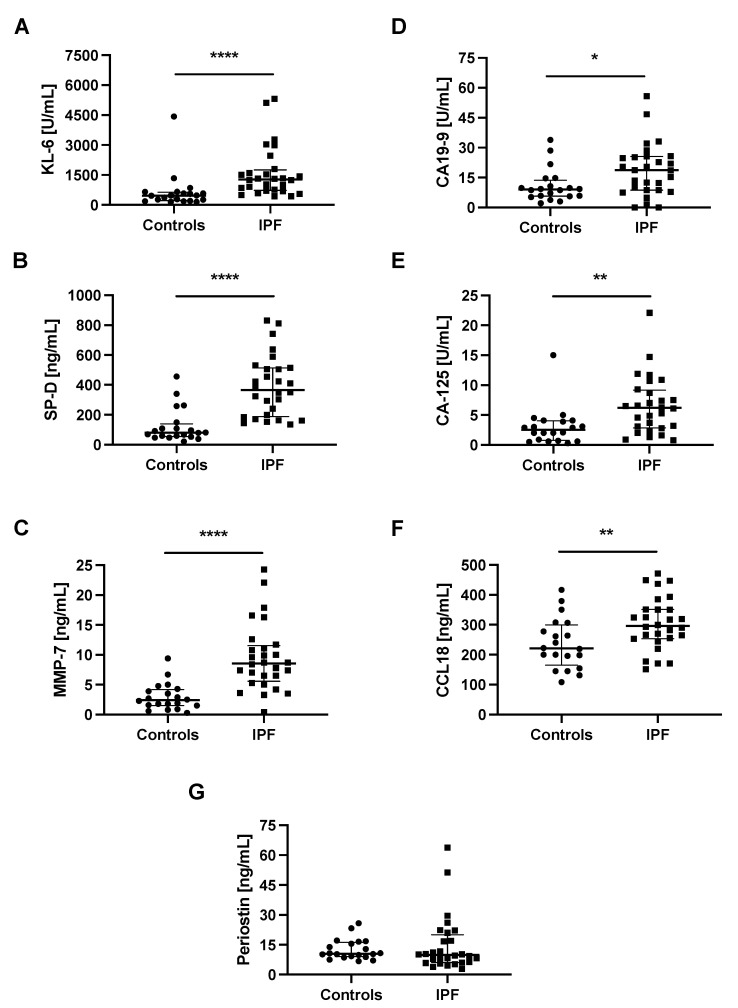
Baseline serum biomarkers levels in patients with IPF and control subjects. Notes: Panels showing baseline serum concentrations of (**A**) KL-6, (**B**) SP-D, (**C**) MMP-7, (**D**) CA19-9, (**E**) CA-125, (**F**) CCL18, (**G**) periostin. * *p* < 0.05; ** *p* < 0.01; **** *p* < 0.0001. Abbreviations: KL-6—Krebs von den Lungen-6, SP-D—surfactant protein D, MMP-7—matrix metalloproteinase 7, CA19-9—cancer antigen 19-9, CA-125—cancer antigen 125, CCL18—chemokine (C–C motif) ligand 18, IPF—idiopathic pulmonary fibrosis.

**Figure 2 jcm-10-03864-f002:**
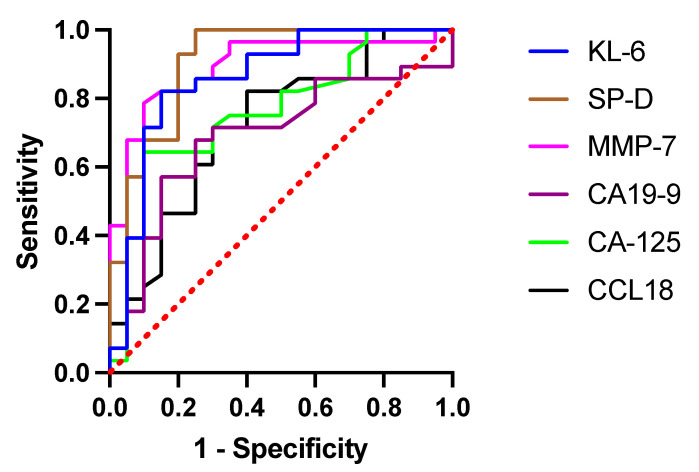
ROC curve analysis in circulating biomarkers to distinguish IPF from control subjects. Abbreviations: ROC—receiver operating characteristic, KL-6—Krebs von den Lungen-6, SP-D—surfactant protein D, MMP-7—matrix metalloproteinase 7, CA19-9—cancer antigen 19-9, CA-125—cancer antigen 125, CCL18—chemokine (C-C motif) ligand 18.

**Figure 3 jcm-10-03864-f003:**
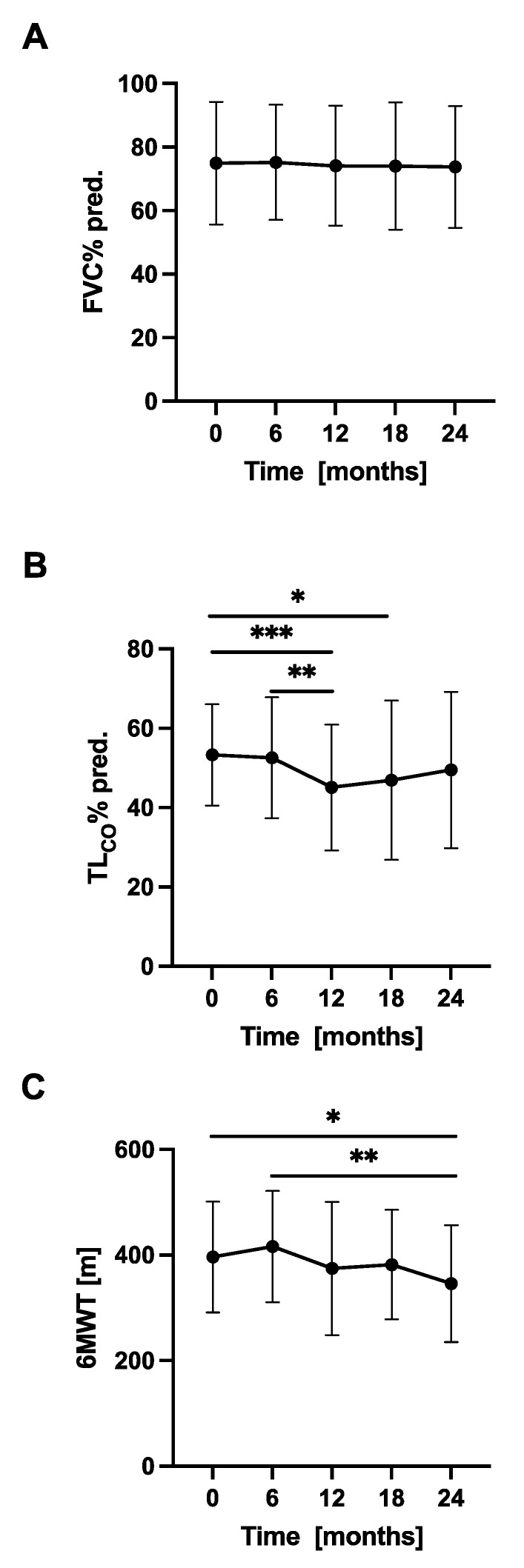
Longitudinal changes in PFTs and 6MWT in patients with IPF. Notes: Panels showing dynamic changes in (**A**) FVC% pred., (**B**) T_L,CO_% pred., (**C**) 6MWT. * *p* < 0.05; ** *p* < 0.01; *** *p* < 0.001. Abbreviations: IPF—idiopathic pulmonary fibrosis, PFTs—pulmonary function tests, FVC—forced vital capacity, T_L,CO_—transfer factor of the lung for carbon monoxide, 6MWT—six-minute walk test.

**Figure 4 jcm-10-03864-f004:**
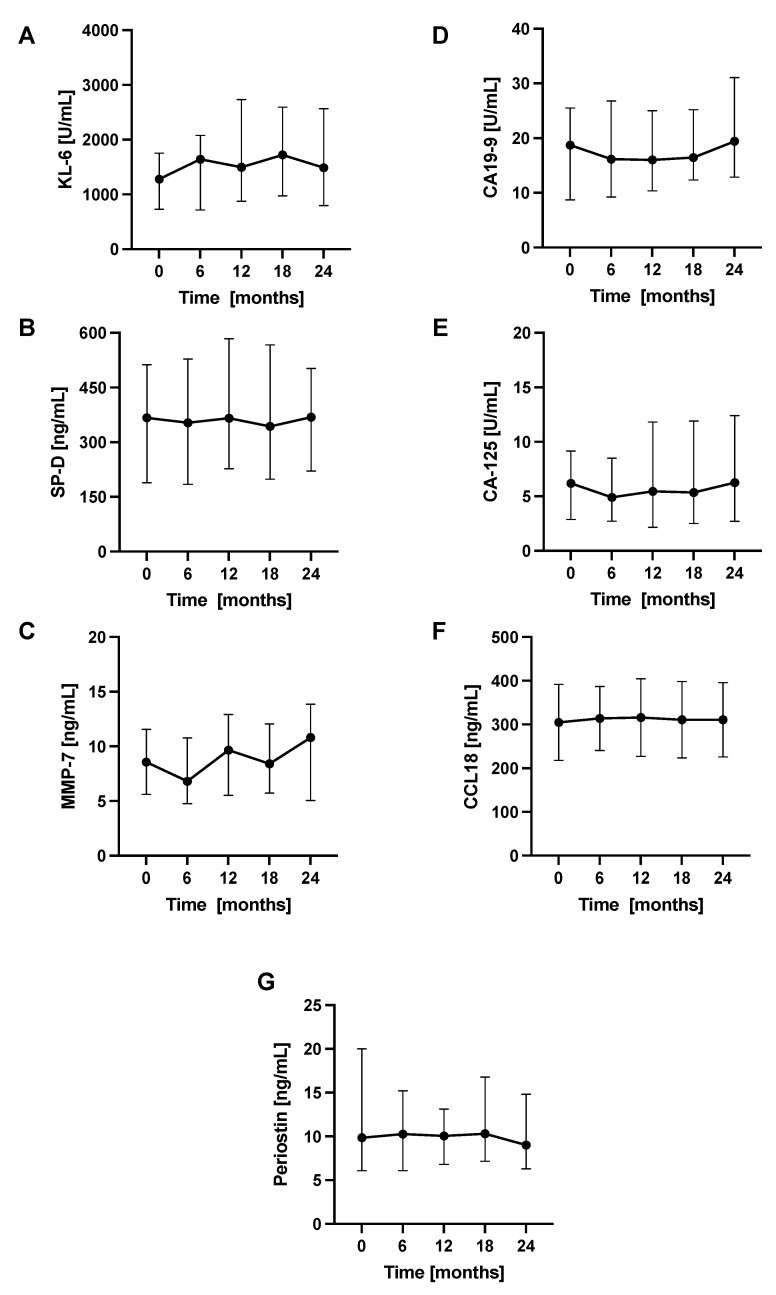
Longitudinal changes in circulating biomarkers concentrations in patients with IPF. Notes: Panels showing dynamic changes in serum concentrations of (**A**) KL-6, (**B**) SP-D, (**C**) MMP-7, (**D**) CA19-9, (**E**) CA-125, (**F**) CCL18, (**G**) periostin. Abbreviations: KL-6—Krebs von den Lungen-6, SP-D—surfactant protein D, MMP-7—matrix metalloproteinase 7, CA19-9—cancer antigen 19-9, CA-125—cancer antigen 125, CCL18—chemokine (C-C motif) ligand 18, IPF—idiopathic pulmonary fibrosis.

**Figure 5 jcm-10-03864-f005:**
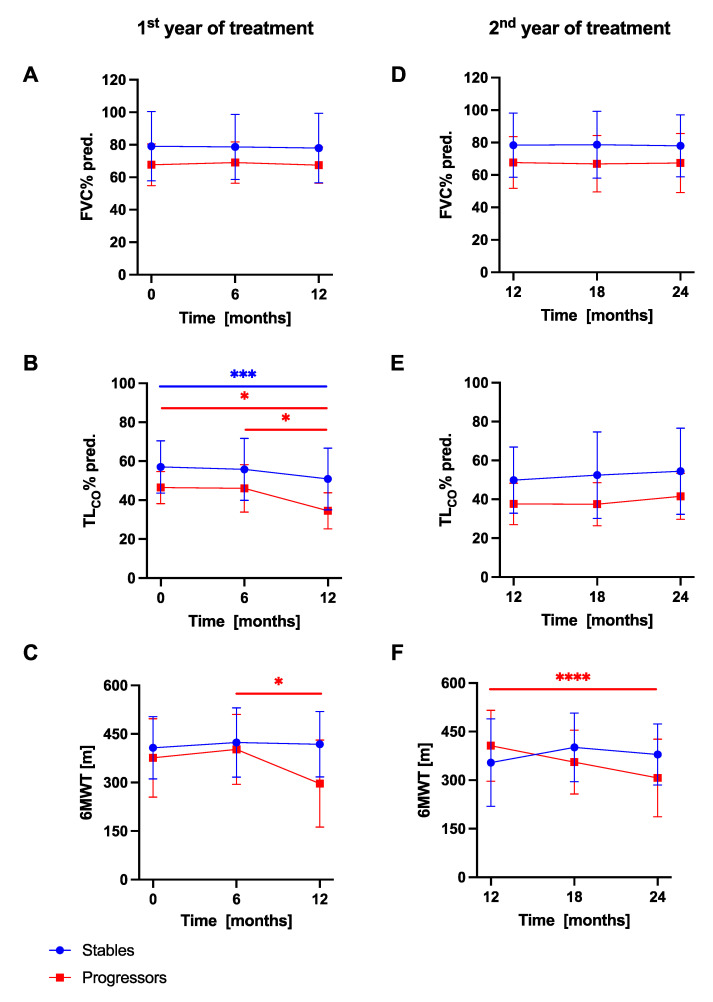
Longitudinal changes in PFTs and 6MWT in IPF patients with stable and progressive disease over the first and the second year of anti-fibrotic therapy. Notes: Panels showing dynamic changes in (**A**) FVC% pred. in the first year, (**B**) T_L,CO_% pred. in the first year, (**C**) 6MWT in the first year, (**D**) FVC% pred. in the second year, (**E**) T_L,CO_% pred. in the second year, (**F**) 6MWT in the second year. * *p* < 0.05; *** *p* < 0.001; **** *p* < 0.0001. Abbreviations: IPF—idiopathic pulmonary fibrosis, PFTs—pulmonary function tests, FVC—forced vital capacity, T_L,CO_—transfer factor of the lung for carbon monoxide, 6MWT—six-minute walk test.

**Figure 6 jcm-10-03864-f006:**
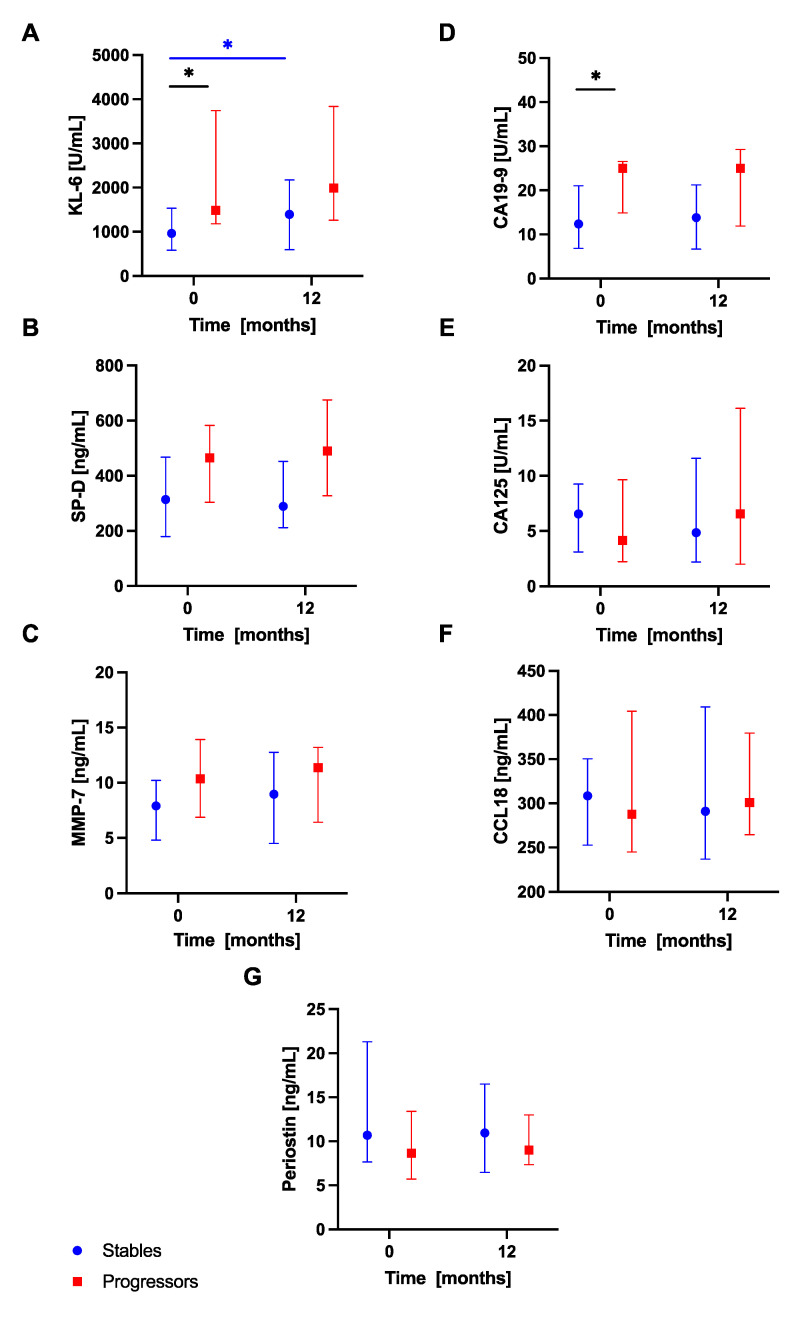
Serum biomarkers concentrations in IPF patients with stable and progressive disease over the first year of anti-fibrotic therapy. Notes: Panels showing serum biomarkers concentrations measured at baseline and 12-month timepoint for: (**A**) KL-6, (**B**) SP-D, (**C**) MMP-7, (**D**) CA19-9, (**E**) CA-125, (**F**) CCL18, (**G**) periostin. * *p* < 0.05. Abbreviations: KL-6—Krebs von den Lungen-6, SP-D—surfactant protein D, MMP-7—matrix metalloproteinase 7, CA19-9—cancer antigen 19-9, CA-125—cancer antigen 125, CCL18—chemokine (C-C motif) ligand 18, IPF—idiopathic pulmonary fibrosis.

**Figure 7 jcm-10-03864-f007:**
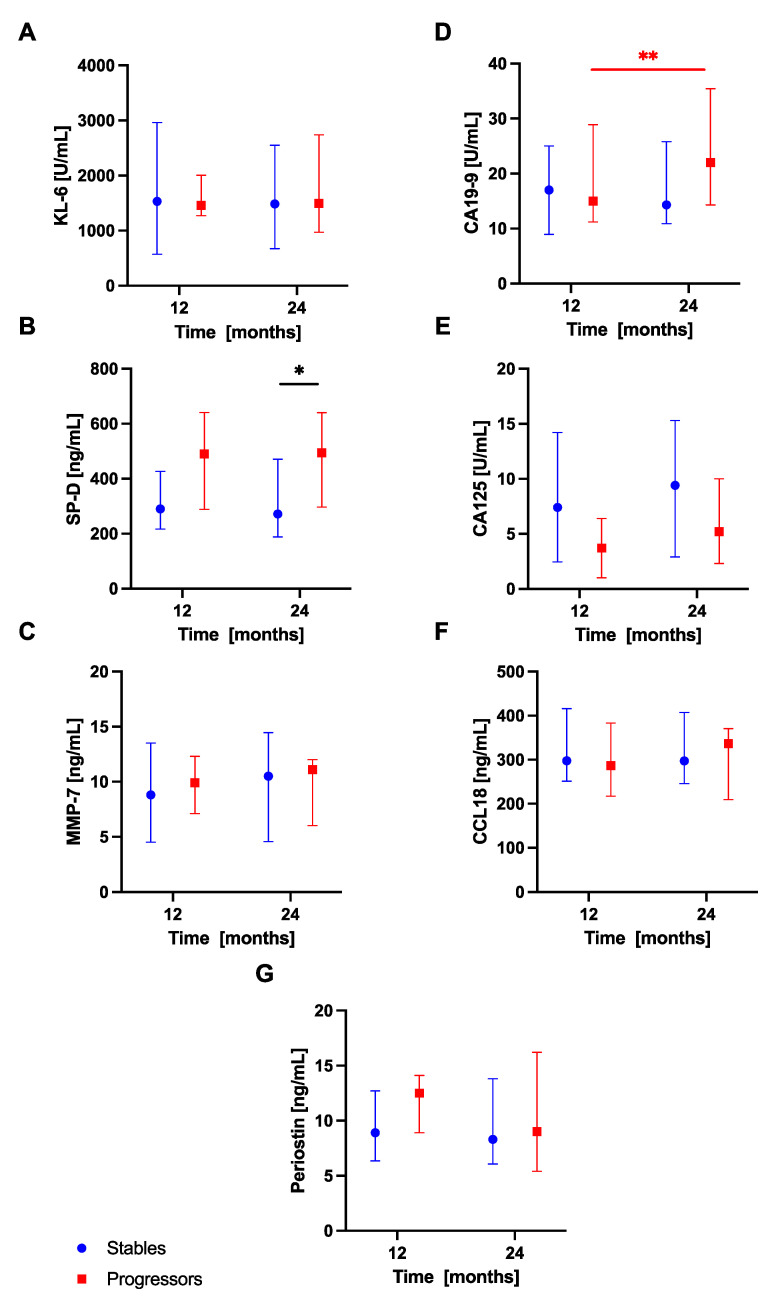
Serum biomarkers concentrations in IPF patients with stable and progressive disease over the second year of anti-fibrotic therapy. Notes: Panels showing serum biomarkers concentrations measured at 12-month and 24-month timepoints for: (**A**) KL-6, (**B**) SP-D, (**C**) MMP-7, (**D**) CA19-9, (**E**) CA-125, (**F**) CCL18, (**G**) periostin. * *p* < 0.05; ** *p* < 0.01. Abbreviations: KL-6—Krebs von den Lungen-6, SP-D—surfactant protein D, MMP-7—matrix metalloproteinase 7, CA19-9—cancer antigen 19-9, CA-125—cancer antigen 125, CCL18—chemokine (C-C motif) ligand 18, IPF—idiopathic pulmonary fibrosis.

**Figure 8 jcm-10-03864-f008:**
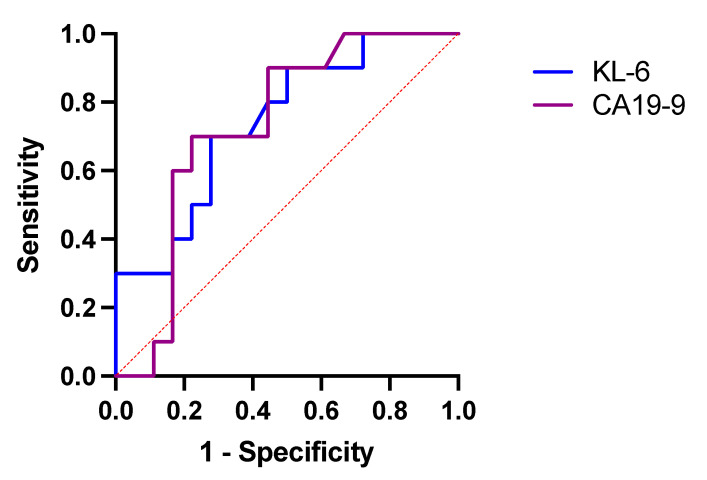
ROC curve analysis in circulating biomarkers to distinguish IPF patients with stable disease from IPF patients with progressive disease over the first year of study follow-up. Abbreviations: ROC—receiver operating characteristic, KL-6—Krebs von den Lungen-6, CA19-9—cancer antigen 19-9.

**Figure 9 jcm-10-03864-f009:**
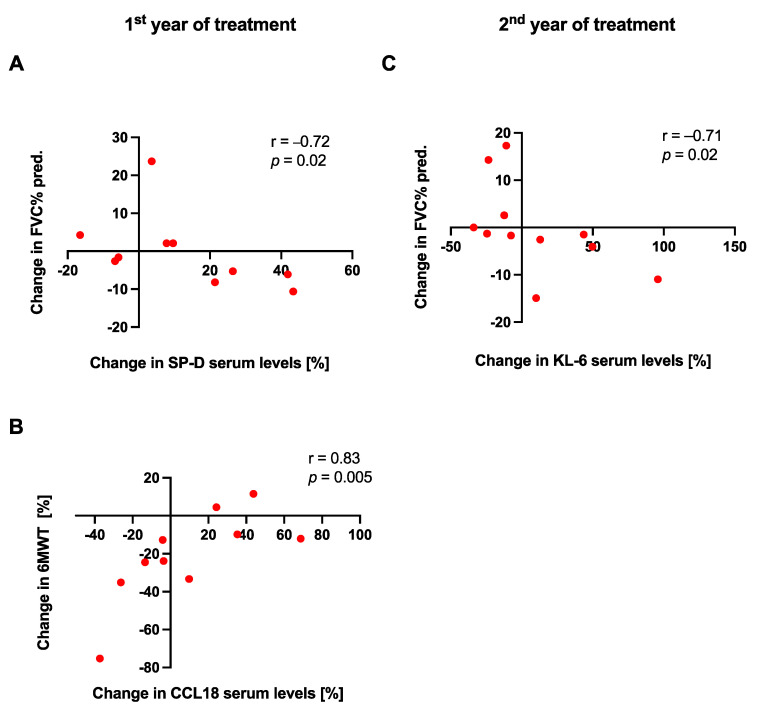
Associations of dynamic changes in serum biomarkers levels with changes in physiologic and functional measures in the progressors subgroup of patients with IPF over the first and second year of anti-fibrotic treatment. Notes: Panels showing a correlation between (**A**) a change in FVC% pred. and a change in SP-D serum levels in the first year of anti-fibrotic therapy; (**B**) a change in 6MWT distance and a change in CCL18 serum levels in the first year of anti-fibrotic therapy; (**C**) a change in FVC% pred. and a change in KL-6 serum levels in the second year of anti-fibrotic therapy; Abbreviations: FVC—forced vital capacity, 6MWT—six-minute walk test, SP-D—surfactant protein D, KL-6—Krebs von den Lungen-6, CCL18—chemokine (C-C motif) ligand 18, IPF—idiopathic pulmonary fibrosis.

**Table 1 jcm-10-03864-t001:** Baseline characteristics of study participants.

	Controls	IPF
Number of subjects	20	28
Sex (male/female)	10/10	17/11
Age (years), mean (SD)	68.40 (6.11)	69.14 (7.86)
Smoking exposure (pack-years) median (IQR)	2.55 (0–36.13)	22.50 (0–30)
Smoking status		
never smokers, *n* (%)	9 (45.00%)	8 (28.57%)
ex-smokers, *n* (%)	3 (15.00%)	19 (67.86%)
current smokers, *n* (%)	8 (40.00%)	1 (3.57%)
GAP index, median (IQR)	N/A	3.0 (3.0–4.0)
GAP stage:	N/A	
Stage I, *n* (%)		17 (60.71%)
Stage II, *n* (%)		11 (39.29%)
Stage III, *n* (%)		0 (0%)
CPI score, mean (SD)	N/A	69.25 (6.87)
Time since diagnosis (years), median (IQR)	N/A	1.79 (0.81–3.73)
FEV_1_ (l), mean (SD)	2.88 (0.81)	2.13 (0.60) ***
FEV_1_ (% of predicted), mean (SD)	106.80 (16.18)	78.95 (19.04) ****
FVC (l), mean (SD)	3.88 (1.21)	2.65 (0.85) ***
FVC (% of predicted), mean (SD)	111.30 (20.73)	74.95 (19.28) ****
FEV_1_/FVC%, mean (SD)	74.15 (6.95)	81.53 (6.94) ***
T_L,CO_ (mmol/min/kPa), mean (SD)	N/A	3.98 (1.08)
T_L,CO_ (% of predicted), mean (SD)	N/A	53.28 (12.78)
6MWT (meters), mean (SD)	N/A	396.30 (104.80)
KL-6 (U/mL), median (IQR)	464.1 (221.4–635.9)	1277.00 (727.8–1755) ****
SP-D (ng/mL), median (IQR)	80.60 (54.35–139)	366.70 (188.8–512.7) ****
MMP-7 (ng/mL), median (IQR)	2.40 (1.53–4.20)	8.55 (5.60–11.55) ****
CA19-9 (U/mL), median (IQR)	9.00 (5.68–13.63)	18.70 (8.70–25.50) *
CA-125 (U/mL), median (IQR)	2.50 (0.75–4.05)	6.20 (2.88–9.15) **
CCL18 (ng/mL), mean (SD)	236.50 (84.69)	304.70 (87.15) **
Periostin (ng/mL), median (IQR)	10.55 (9.00–16.25)	9.85 (6.10–20.00)

**Notes:** * *p* < 0.05; ** *p* < 0.01; *** *p* < 0.001; **** *p* < 0.0001. Abbreviations: IPF—idiopathic pulmonary fibrosis, FEV_1_—expiratory volume in 1 s, FVC—forced vital capacity, T_L,CO_—transfer factor of the lung for carbon monoxide, 6MWT—six-minute walk test, GAP—gender-age-physiology, CPI—composite physiologic index, KL-6, Krebs von den Lungen-6; SP-D, surfactant protein D; MMP-7, matrix metalloproteinase 7; CA19-9, cancer antigen 19-9; CA-125, cancer antigen 125; CCL18, chemokine (C-C motif) ligand 18.

**Table 2 jcm-10-03864-t002:** Discriminatory ability and cut-off values of circulating biomarkers by ROC curve analysis distinguishing IPF from controls.

	KL-6	SP-D	MMP-7	CA19-9	CA-125	CCL18
AUC	0.8589	0.9089	0.8946	0.6893	0.7723	0.7232
95% CI	0.7430–0.9749	0.8196–0.9983	0.8011–0.9882	0.5341–0.8445	0.6355–0.9091	0.5750–0.8715
*p*-value	<0.0001	<0.0001	<0.0001	0.0266	0.0014	0.0089
Cut-off value	673.95 U/mL	121.80 ng/mL	5.15 ng.mL	11.50 U/mL	4.55 U/mL	242.05 ng/mL
Sensitivity	82.14%	100.00%	78.6%	67.86%	64.29%	82.14%
Specificity	85.00%	75.00%	90.00%	75.00%	90.00%	60.00%

Abbreviations: ROC—receiver operating characteristic, KL-6—Krebs von den Lungen-6, SP-D—surfactant protein D, MMP-7—matrix metalloproteinase 7, CA19-9—cancer antigen 19-9, CA-125—cancer antigen 125, CCL18—chemokine (C-C motif) ligand 18, AUC—area under ROC curve, CI—confidence interval.

**Table 3 jcm-10-03864-t003:** Discriminatory ability and cut-off values of circulating biomarkers by ROC curve analysis distinguishing stables from progressors over the first year of study follow-up.

	KL-6	CA19-9
AUC	0.7417	0.7306
95% CI	0.5535–0.9298	0.5402–0.9209
*p*-value	0.0370	0.0466
Cut-off value	1328 U/mL	21.20 U/mL
Sensitivity	70.00%	70.00%
Specificity	72.22%	77.78%

## Data Availability

The datasets analyzed during a current study are available on request from the corresponding author on reasonable request.

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
