# Peer review of "Serial Measurements of Circulating KL-6, SP-D, MMP-7, CA19-9, CA-125, CCL18, and Periostin in Patients with Idiopathic Pulmonary Fibrosis Receiving Antifibrotic Therapy: An Exploratory Study"

_jcm, 2021, doi:10.3390/jcm10173864_

Round 1

Reviewer 1 Report

In this paper, the authors deal with serum biomarkers in patient with IPF receiving antifibrotic agents. 

Major comments for Author:

  1. The reviewer thinks that the validity of the IPF progression is not explained sufficiently. Phase 3 trial of pirfenidone (ASCEND study) and phase 3 trials of nintedanib (INPULSIS-1 and INPULSIS-2) used absolute change of the percentage of the predicted FVC. Moreover, du Bois et al (2011) suggested that the MCID of 24-week absolute change in percent-predicted FVC was 2-6%. The authors should give a reason why they choose relative decline of FVC% predict in their composite definition of IPF progression. Thus, the description “50 meters relative decline in 6MWT” should be changed.
  2. Ohta S et al (2017) showed that monomeric periostin is associated with both declines of pulmonary function and overall survival in patient with IPF. Authors need to confirm the periostin assay kit detect whether total periostin or monomeric periostin.

Minor Comment:

Section number should be 3.5. in Line 278.

Reviewer 2 Report

Review comments

General comments

In the present study, the authors showed that the well-established markers, KL-6, SP-D, and MMP-7 and more recently introduced one, CA19-9, are useful to distinguish healthy and IPF patients using ROC analysis. Also, the authors showed the longitudinal changes of serum biomarkers in the stable and progressive IPF patients. The authors performed many statistical analyses, prepared many figures, and discussed the results with the previously reported papers in the clear English-writing.

However, there are several issues which should be addressed for the publication. First, it is difficult to find the novelty of this study especially in the first section using KL-6, SP-D, and MMP-7, which are well-established markers in IPF. Second, the study design and the interpretation of the results was not clear enough. Third, the number of the patients for the study was too small.

Major comments

  1. In Figure 1, the authors confirmed that the established serum markers are significantly different from controls to IPF. The purpose to perform this analysis was unclear. Further, in Table 2, the authors perform ROC curve analysis even in the Periostin, which did not show the significant different between IPF and healthy controls. The authors should explain the reason why this analysis was needed.
  2. In Table 3, the authors showed the changes of several data within 24 months in IPF patients. However, considering the later analysis, it is not clear why the authors did not divide the data into stable and progressive IPF. Moreover, it is more understandable to present the results with graphs (with each data point) than the table.
  3. In Figure 3 and 4, the authors showed the concentrations of serum biomarkers in stable and progressive IPF. To my eye, the data are the similar in Figure 3 to Figure 5-6 and Figure 4 to Figure 7-8. Why did the authors not combine those data in the one Figure? The authors describes that there is significant different in CA19-9 in progressive IPF group within 12 to 24 months in Figure 8D. However, it is not shown in Figure 4D. The same pattern is Figure 5A and Figure 3A.
  4. Figure 6 shows that there is no clear tendency of CCL18 in the progressive disease. Nevertheless, the authors show that the changes in CCL18 was associated with the changes in 6MWT. It is unclear why only 6MWT showed such a pattern if the other PFT markers did not?
  5. Why did the authors not try to analyze with the ROC curve to distinguish these stable and progressive IPF patients?
